# Transcriptional Targeting Approaches in Cardiac Gene Transfer Using AAV Vectors

**DOI:** 10.3390/pathogens12111301

**Published:** 2023-10-30

**Authors:** Lena C. Schröder, Derk Frank, Oliver J. Müller

**Affiliations:** 1Department of Internal Medicine III, University Hospital Schleswig-Holstein, Campus Kiel, 24105 Kiel, Germany; lena.schroeder2@uksh.de (L.C.S.); derk.frank@uksh.de (D.F.); 2DZHK (German Centre for Cardiovascular Research), Partner Site Hamburg/Kiel/Lübeck, 24105 Kiel, Germany

**Keywords:** gene therapy, cardiac promoters, cardiac enhancers, adeno-associated virus (AAV) vectors, transcriptional targeting

## Abstract

Cardiac-targeted transgene delivery offers new treatment opportunities for cardiovascular diseases, which massively contribute to global mortality. Restricted gene transfer to cardiac tissue might protect extracardiac organs from potential side-effects. This could be mediated by using cis-regulatory elements, including promoters and enhancers that act on the transcriptional level. Here, we discuss examples of tissue-specific promoters for targeted transcription in myocytes, cardiomyocytes, and chamber-specific cardiomyocytes. Some promotors are induced at pathological states, suggesting a potential use as “induction-by-disease switches” in gene therapy. Recent developments have resulted in the identification of novel enhancer-elements that could further pave the way for future refinement of transcriptional targeting, for example, into the cardiac conduction system.

## 1. Introduction

Gene therapy could become a promising treatment option for many high-prevalence cardiac diseases like heart failure and atrial fibrillation, but also for rare genetic conditions. Adeno-associated virus (AAV) vectors have emerged as a workhorse for somatic gene therapy with impressive results in clinical trials for inherited amaurosis and spinal muscular atrophy that have led to FDA-approved treatments [1]. With distinct naturally occurring AAV serotypes displaying a tropism for the heart (for example AAV9), AAV-based gene therapies are currently being studied for several cardiac diseases [1]. However, cardiac gene therapy is challenged by potential off-target effects that potentially threaten other organs. To overcome this limitation, gene expression restricted to the heart or even diseased myocardium could optimize treatment and avoid side effects due to off-target expression. Specificity can be improved through targeting strategies on the transductional, transcriptional, or posttranscriptional level. Transduction targeting of the heart takes advantage of distinct modifications of the AAV capsid surface and is reviewed elsewhere [2]. Posttranscriptional (de)targeting could be achieved by including three repeats of microRNA (miR)122 target sites into the 3′UTR of an AAV vector genome. Intravenous injection of AAV vectors with such de-targeted genomes into adult mice resulted in efficient suppression of hepatic gene expression in mice expressing high amounts of miR122 in the liver [3]. In contrast, transgene expression in the heart remained high as murine hearts express minimal amounts of miR122. However, this strategy requires high expression levels of miR122 in the organs to be de-targeted. As it turned out that hepatic expression of miR122 is highly variable, even between different mouse lines, a clinical translation of such de-targeting approaches remains challenging [4]. In contrast, transcriptional targeting not only allows the suppression of gene expression in extracardiac tissue, but also has the potential to further increase gene expression via utilization of distinct cardiac enhancers. 

Cis-regulatory elements including promoters and enhancers regulate gene transcription. The core promoter is typically located between −50 to +50 bp from the transcription start site (TSS) and initiates transcription by assembling the preinitiation complex [5]. Further regulation of expression is defined by the 50 to 1000 bp proximal promoter, located upstream of the TSS, that carries several unique transcription factor binding sites [6]. Enhancers are the most interesting distal regulatory elements in gene therapy since they can massively increase transcription by binding transcription factors and often provide high cell specificity [7]. This review aims to give an overview of current promoter and enhancer sequences used for transcriptional targeting of AAV vectors into the myocardium and discuss future lines of research.

## 2. Promoters Used for Transcriptional Targeting of AAV Vectors to the Myocardium

The use of cardiac-specific promoters is a promising strategy to increase the specificity of transgene expression for cardiac gene therapy [8]. In comparison to constitutively active promoters like the cytomegalovirus (CMV) promoter, specific promoters allow restriction to distinct cell types [8]. Nowadays, several myocardial promotors predominantly active in cardiomyocytes are used in preclinical studies, but further levels of specificity could improve therapeutical approaches, e.g., for cardiac arrhythmia [3,9,10,11]. On the one hand, specificity can be increased by driving transgene expression specifically in cardiomyocyte subtypes, for example atrial, ventricular, and nodal cells, or distinct pathologies to avoid side effects in unaffected tissue (e.g., targeting of ischemic myocardium [12]). On the other hand, broader promoter activity is required when, in addition to cardiomyocytes, skeletal muscles should be targeted, for example in muscular dystrophies affecting both skeletal and cardiac myocytes [13] (Figure 1).

Cardiomyocytes can be distinguished into contractile and non-contractile types [14]. Additionally, they are characterized according to chamber-specific differences. Contractile cardiomyocytes comprise the myocardium but vary in ultrastructure, gene expression patterns, and function according to their location in the chambers. Thus, atrial natriuretic factor (ANF) and sarcolipin (SLN) are exclusively expressed in atrial cardiomyocytes and mediate the control of blood pressure, salt balance, and calcium [15,16]. Concomitantly, ventricular cardiomyocytes are characterized by myosin light chain (MLC-2v) expression, which is involved in forming cardiac sarcomeres and maintaining ventricular contractility [17]. The cardiac troponin T (cTnT) and alpha myosin heavy chain (αMHC) genes are expressed in both atrial and ventricular cardiomyocytes and are involved in the contractile function [18,19]. Specialized non-contractile cardiac cells generate and transduce electrical signals to maintain heart muscle contraction. Collectively, these cells are considered the cardiac conduction system (CSS) and are distinguished into the sinoatrial-node (SAN), atrioventricular-node (AVN), Bachmann bundle (BB), and Purkinje fibers [20].

The following paragraphs discusses the advantages and disadvantages of different promoters for cardiac gene transfer using AAV vectors.

### 2.1. Myosin Light Chain (MLC)-2v Promoter

Myosin light chain 2v is both expressed in myocardium and slow-twitch skeletal muscle [19]. Initial studies in targeting cardiac gene therapy used a 2.1 MLC-2v promoter for transgenic expression from adenoviral vectors [21]. They showed predominant reporter gene expression in the ventricular myocardium of neonatal rats after local injection in the cardiac cavity, although the vector system itself transduced other non-target organs. Unfortunately, the MLC-2v promoter is highly active in developmental stages but downregulated in adult myocardium, leading to low promotor activity and insufficient transgene expression [22]. Fusion of a CMV enhancer element to a 1.5 kb MLC-2v promoter fragment restored gene expression in left and right ventricular myocardium in adult mice upon intravenous injections of AAV6 vectors with lower expression in atria and skeletal muscle [23]. The skeletal muscle activity was not surprising when considering the expression pattern of endogenous myosin light chain-2v, which is expressed both in cardiac and fast-twitch skeletal muscle fibers [22]. A similar skeletal activity was also observed in transgenic mice harboring a CMVenh/MLC2.1 luciferase transgene [23]. The CMV-enhanced 1.5 MLC-2v promoter also enabled cardiac-restricted gene expression in pigs upon application of AAV6 vectors via coronary venous retroinfusion [24]. Transcriptional targeting with the CMV-enhanced 1.5 MLC-2v promoter allows only approximately 1.5 kb space for transgene cDNAs in AAVs in a therapeutical context due to the AAV genome packaging limit. Moreover, it cannot be used for the generation of self-complementary AAV vectors, which require even smaller vector genomes. Thus, shorter but still specific promoter sequences would be advantageous. Further shortening of the MLC-2v promoter resulted in a 263 bp MLC-2v promoter fragment coupled to the CMV enhancer [25]. However, systemic delivery of AAV9 vectors with this CMV-0.26MLC-2v promoter in mice resulted in significant background expression in the liver [3]. Nevertheless, the expression activity in the heart of the truncated MLC-2v hybrid promoter was comparable to the high-expressing CMV promoter. This small CMV-0.26MLC-2v hybrid-promoter construct was successfully utilized for cardiac gene transfer in preclinical large animal studies for gene therapy of heart failure with AAV6 and AAV9 vectors [11,26]. The absence of significant extracardiac gene expression in these studies might be explained by a combination of locoregional cardiac application using coronary venous retroinfusion and the higher specificity of the CMV-0.26MLC-2v hybrid-promoter compared to unspecific promoters. Interestingly, a shortened 263 bp MLC-2v promoter fragment without an additional enhancer was used within lentiviral vectors and provided stable transgene expression not only in the heart but in the liver and spleen as well [27,28]. In contrast to Geisler et al. [3], Phillips et al. showed that the fusion of hypoxia-response elements (HPE) to the short 260bp promoter fragment allows the generation of a hypoxia “vigilant” vector [29]. AAV vectors harboring a vascular endothelial growth factor (VEGF) cDNA driven by such a short MLC-2v promoter fused to nine copies of a hypoxia-response element (HRE) resulted in significantly higher VEGF expression in ischemic myocardium of mice than in non-ischemic hearts, without detection of any VEGF in livers [30]. In conclusion, the MLC-2v promoter mediates efficient cardiac transgene expression, in particular when coupled to a CMV enhancer element. The latter compensates the low activity of the MLC-2v promoter due to its developmental downregulation in adult tissue. On the other hand, the constitutively active CMV enhancer might contribute to off-target expression after systemic delivery of the vector. MLC-2v promoter activity was detectable even under pathological conditions, like hypoxia, which could be essential for clinical applications. So far, it is challenging to maintain heart-specificity and high transgene expression under a shortened MLC-2v promoter.

### 2.2. Atrial Natriuretic Factor (ANF) Promoter

In early developmental stages, ANF is ubiquitously expressed in the heart but restricted to atrial chambers in the postnatal myocardium [31]. In hypertrophied and failing hearts, the ANF expression is strongly upregulated and extends to ventricular tissue, suggesting the ANF promoter as a potential ‘induction-by-disease’ switch in heart failure [32]. Systemic injection of an AAV9 vector harboring a 653 bp ANF promoter fragment enables significant expression of a GFP reporter in murine atria [33]. The same vector was used to introduce an atria-specific liver kinase-B1(LKB1) knockdown in LKB1^fl/fl^ mice via delivery of Cre [34]. Fusion of a 530 bp CMV enhancer element to the 564 bp (−473 to +91) ANF promoter in a lentivirus increased GFP expression in an atrial cardiomyocyte cell line (HL-1) over the ANF promoter alone [27]. Expression in off-target fibroblasts remained negligible, despite constitutive CMV enhancer activity, and the validation in vivo remained open. Although ANF expression is upregulated in the whole myocardium as part of the hypertrophic response mechanisms [35], in vivo studies in transgenic mice did not reveal ANF promoter activity in the ventricular myocardium of hypertrophic hearts generated via pressure overload [33,35]. Surprisingly, atria-specificity was preserved upon induction of heart failure via pressure overload (transverse aortic constriction, TAC) after AAV9-mediated systemic gene transfer of a Cre-recombinase under control of the ANF promoter in Tomato reporter mice, a transgenic line enabling sensitive tracing of Cre expression using a red-green fluorescence reporter switch [33]. These in vivo results indicate an atria-restricted activity of the ANF promoter under hypertrophic conditions and discourage using the ANF promoter for gene therapy of heart failure. It is still not finally resolved whether lacking ventricular expression is limited to hypertrophy that is experimentally induced via TAC and may differ in other heart failure models or patients. Another possibility is that the ANF promoter fragments tested so far lack specific regions for expression in failing ventricular myocardium. Taken together, a relatively short ANF promoter fragment mediates stable and specific transgene expression in atrial cardiomyocytes delivered via different vector systems in vitro and in vivo. This paves the way for future clinical approaches to target highly prevalent diseases of the atria.

### 2.3. Sarcolipin (SLN) Promoter

Sarcolipin is predominantly expressed in atrial myocardium and skeletal muscle, but not ventricular myocardium [15]. The 1029 bp sarcolipin promotor enabled a very low reporter gene activity in the atrium compared to the constitutively active CMV promoter after systemic delivery through AAV9 in mice. The SLN promoter revealed significant activity in ventricular myocardium, skeletal muscle, and the liver [36]. To overcome the limited activity of the SLN promoter, Yoo et al. fused 1029 bp of the SLN promoter to the cardiac specific cis-regulatory module 4 (CS-CRM4) from the human calsequestrin 2 gene [37]. The addition of CS-CRM4 improved the expression levels and atrial specificity of the SLN promoter in mice after systemic administration. The SLN expression varies in several cardiac diseases and is upregulated in congenital heart disease, preserved ejection fraction, and mitral regurgitation [38,39]. Thus, increased SLN promoter activity under these pathological states could facilitate disease-induced transgene expression and pave the way for gene therapy of the atrium. However, in atrial fibrillation, reduced SLN expression might have a negative effect on promoter activity. Thus, the SLN promoter might be unsuitable, and an ANF-promoter-based approach could be more advantageous for atria-specific gene therapy targeting atrial fibrillation. A further limitation of the SLN-CS-CRM4 hybrid promoter is its rather large size (1496 bp), limiting the packaging capacity of AAV vectors.

### 2.4. Cardiac Troponin T (cTnT) Promoter

The cardiac troponin T (cTnT) isoform is encoded by the TNNT2 gene and expressed in cardiac muscle and, transiently, also in embryonic and neonatal skeletal muscles, including both slow- and fast-fiber-dominant muscles in avians and humans [40,41]. Prasad et al. showed almost exclusive and substantial GFP or luciferase expression in the myocardium of young mice under the further truncated 418 bp (−375 to +43) chicken cTnT promoter upon systemic injection of AAV8 and AAV9 vectors [42]. The promoter activity was only about 40% of expression levels controlled by the CMV promoter. To improve its activity, the chicken cTnT promoter construct was fused to the synthetic cardiac CS-CRM4 enhancer element, resulting in a 637 bp hybrid promoter [9]. After systemic application of an AAV9 vector, the hybrid promoter could achieve stable and excessive gene expression in murine myocardium at levels comparable to the CMV promoter. Off-target expression tended to zero; thus, adding the CS-CRM4 enhancer element further improved the cardio-specificity of the chicken cTnT promoter. The short size allows much space for the cDNA of a therapeutic transgene, making the cTnT-CS-CRM4 hybrid promoter a promising tool for AAV-mediated cardiac gene transfer.

In addition to the avian, the human cTnT promoter is also a promising tool for future clinical applications in gene therapy or preclinical studies. Werfel et al. transferred the Cre recombinase gene under the control of a 544 bp human cTnT promoter element systemically delivered with an AAV9 vector in a rosa26-lacZ reporter mouse line [10]. Cre recombinase activity confirmed a highly cardiac-specific gene expression with only very little off-target expression. Moreover, the 544 bp human cTnT-promoter targeted a luciferase reporter gene into murine myocardium with an efficiency comparable to the CMV promoter.

All in all, the human or chicken cTnT promoters mediate stable and specific transgene expression in vivo that—at least in case of the chicken promoter—may be further improved by a synthetic enhancer element. The cTNT promoter is advantageous for use in AAVs due to the small fragment size that is sufficient to maintain specific activity in the myocardium. An exemplary therapeutical approach for the cTnT promoter in cardiac gene therapy was introduced by Bezzerides et al. [43]. Catecholaminergic polymorphic ventricular tachycardia (CPVT) is an inherited form of cardiac arrhythmia, where CaMKII contributes to arrhythmogenesis [44]. Cardiomyocyte-specific inhibition of CaMKII brings up a potential therapeutical target in a gene therapeutical context. Expression of the CaMKII inhibitory peptide (AIP) under the control of a human cTnT promoter was integrated into an AAV9 vector. In a murine CPVT model, systemic application of the vector provided myocardial-specific transgene expression with only minimal extra-cardiac expression and suppression of arrhythmia. In contrast to other tissue-specific promoters like ANF or SLN, transgene expression mediated by the cTnT promoter alone (without enhancement) appears adequate. Since CPVT arrhythmias exclusively affect the ventricular cardiomyocytes, using a chamber-specific and not myocardial-specific promoter would protect unaffected atrial tissue in patients.

The cTnT promoter has been applied in many further AAV-based approaches of cardiac gene transfer into rodents [45,46]. Further work is necessary to validate the cTNT promoter in large animal models.

### 2.5. Alpha Myosin Heavy Chain (αMHC) Promoter

Early studies tested a 1 kb (−612 to +420) αMHC promoter packed in an adenoviral vector for atria-specific gene expression [22]. Injection of the recombinant adenovirus into the cardiac cavity of neonatal mice revealed a promoter activity in the whole heart and also in off-target organs like the lung and liver, and transgene expression was detected. The reason is that αMHC expression is atria-specific in embryonal hearts but extends to the ventricular tissue after birth, making the promoter activity heart- but not chamber-specific [47]. Low cardiac expression was mediated by a 363 bp (−344 to +19) αMHC promoter fragment after AAV-mediated transduction in vivo [47]. The expression level was lower compared to the CMV promoter, but lacking expression in off-target cells indicated a cardio-specific αMHC promoter activity after local administration. The truncated αMHC promoter could be improved by the addition of enhancer elements. In the studies of Rincon et al., GFP reporter expression was achieved in murine cardiomyocytes after systemic delivery of an AAV9 vector carrying a αMHC-CS-CRM4 hybrid promoter [37]. The expression level driven by the hybrid promoter was 100-fold higher than with the αMHC promoter alone. The synthetic enhancer element increased not only promoter activity but also heart specificity, making the hybrid-promoter construct a promising tool in gene therapy for genetic heart conditions. A preclinical study introduced a potential therapeutical lentivirus to target Fabry disease, a genetic lysosomal storage disease where Gb3 accumulates in the heart and other organs [48]. A 1.2 kb (−1198 to +1) αMHC promoter fragment controlling luciferase expression showed cardiac-restricted activity in lentiviral-transduced mice after systemic delivery [28]. Luciferase expression was more intense when controlled by cTnT and EF1α promoters, but the αMHC promoter provided the highest level of specificity in the lentiviral context. In a Fabry mouse model, application of the therapeutical vector harboring the αMHC promoter led to a significant reduction of globotriaosylceramide (Gb3) in the heart. The lentivirus carrying α-galA cDNA under MHC promoter control led to a significant increase in α-galA activity, reducing Gb3 accumulation. However, in failing hearts the MHC expression tilts towards βMHC and not αMHC, leading to reduced αMHC promoter activity [49]. Thus, the αMHC promoter might have disadvantages for therapeutical applications targeting hypertrophic cardiomyocytes.

### 2.6. Muscle Creatine Kinase (MCK) Promoter

The MCK promoter is highly active in all striated muscles, which includes cardiac and skeletal myocytes [50]. Transgenic expression was massively increased in a murine myogenic cell line under the control of a −358 to + 7 CK promoter fragment fused to the 206 bp modified CK enhancer element 2RS5, even compared to the CMV promoter [51]. Intravenous injection of the 571 bp ‘CK6’ expression construct packed in an adenoviral vector reviewed high, muscle-specific, and long-term (4 months) gene expression in a mouse model for Duchenne muscular dystrophy (DMD) with only low ectopic expression in liver. Further promising MCK expression cassettes were introduced by the same lab [51]. MHCK7 is based on the CK6 construct with different modifications and the addition of a 188 bp enhancer sequence of the αMHC gene. Systemically delivered by AAV6, the 770 bp MHCK7 cassette mediates high-level gene expression in skeletal and cardiac myocytes in wild-type mice with little off-target expression [52]. In a murine model of Pompe disease, a lysosomal storage disorder, the MHCK7 promoter restores α-glucosidase expression in heart and skeletal muscle delivered by AAV8 vectors [53].

Stable myocyte-restricted gene expression is a promising tool for gene replacement therapy to target muscular dystrophies. In DMD, X-linked recessive mutations in the dystrophin gene lead to the failing assembly of the dystrophin-glycoprotein complex, resulting in severe cardiac and skeletal muscle wasting. Due to promising results in large animal studies, a clinical trial used an AAV9 vector that delivers microdystrophin under control of a CK8 promoter to boys with Duchenne (NCT03368742). The 837/980 bp CK8 promoter construct is based on MHCK7, but the αMHC enhancer fragment is replaced with two copies of the 206 bp MCK enhancer. Success of clinical trials led to approval of delandistrogene moxeparvovec (ELEVIDYS), a recombinant AAV rh74 vector harboring the MHCK7 promoter element to express micro-dystrophin in pediatric Duchenne patients (NCT03375164). In addition to trials for Duchenne, a gene therapy approach is currently being evaluated in a clinical trial where the MHCK7 promoter drives β-sarcoglycan expression in β-sarcoglycan deficient Limb-Girdle Muscular Dystrophy patients (NCT03652259). The underlying data indicate a successful application of tissue-specific expression cassettes. The variety of thriving preclinical and clinical trials and the approval of one gene therapy vector suggest that MCK expression cassettes may be used as promoters for transcriptional targeting in neuromuscular diseases.

### 2.7. Desmin Promoter

Different combinations of promoter/enhancer sequences of the gene encoding desmin have been recently reviewed in detail [13]. The desmin promoter is active in cardiac, skeletal, and smooth muscles, and in vivo studies indicated similar expression levels mediated by the desmin promoter in comparison to the highly active CMV promoter [13]. Promising preclinical studies revealed, for example, the construct rAAV.DES.hGAA, which drives acid *α*-glucasidase expression under a human desmin promoter-enhancer construct packaged into the AAV9 capsid. The vector is currently being tested in a clinical trial to restore acid *α*-glucasidase levels in patients with Pompe disease for reduction of glycogen accumulation in cardiac and skeletal muscle tissue (NCT02240407).

### 2.8. SPc5-12 Synthetic Promoter

To overcome low levels of transgenic expression in the skeletal and cardiac muscle cells, Li et al. combined myogenic regulatory elements to generate a library of synthetic muscle-specific promoters with strong activity [54]. The most potent SPc5-12 carried a cassette of several transcription factor binding sites (SRE, MEF-2, MEF-1, and TEF-1) coupled to 448 bp (−424 to + 24) skeletal α-actin promoter. SPc5-12 mediates muscle-specific activity with a sixfold increase over the CMV promoter. Luciferase expression was massively increased in immune-deficient mice (SCID) injected with AAV9 vectors harboring a SPc5-12 promoter coupled to the CS-CRM4 enhancer element compared to the CMV promoter (19-fold) or SPc5-12 promoter alone (2-fold) [37]. Expression levels of striated muscles were increased, and a low ectopic expression was observed in the lung. Malerba et al. tested the SPc5-12 CS-CRM4 hybrid construct to restore micro-dystrophin expression in cardiac and skeletal muscle in a Duchenne mouse model after systemic injection [55]. Although many clinical studies for DMD are ongoing, the hybrid construct might be advantageous over the MCK promoter cassettes for DMD-targeted gene therapy, especially in restoring dystrophin levels in cardiac tissue. Heart failure, the major cause of death in DMD patients, could be prevented with a more cardio-directed approach. These studies indicate SPc5-12 as a synthetic promoter providing high muscle-specific transgene expression with high activity in the heart in vivo.

## 3. Cardio-Specific Enhancers

In comparison to ubiquitously active promoters like the CMV promoter, activity of tissue-specific promotors is often very poor [56]. Enhancer sequences could improve expression activity via fusion to the promoter or mediate specificity. Rincon et al. designed novel cardiac-specific cis-acting regulatory modules (CS-CRMs) via computational de novo design [37]. Five of the eight generated CS-CRMs, comprising different transcription factor binding sites of different heart-specific genes, mediated 10-fold increased transgene (GFP) expression in murine myocardium after systemic delivery. The highest cardiac-specific expression was achieved when the AAV9 vector delivered a transgene controlled by CS-CRM4 or CS-CRM7. The CS-CRM4 corresponds to clusters of transcription factor binding sites of the heart-specific calsequestrin 2 gene and has a length of 219 bp. The regulatory element mainly boosts the performance of the αMHC promoter. Combined with other (cardiac)-muscle-specific promoters like SPc5-12, cTnT, MLC2v, and SLN, the CS-CRM4 element predominantly increased the specificity of cardiac gene expression [9,36,37,57]. In a large-scale approach, a library comprising 400 bp sequences of potentially cardio-specific enhancer elements was screened for identification of novel enhancer sequences in mouse hearts in vivo using an AAV9-based assay [58]. The sequences for the enhancer library were chosen according to transcription factor occupancy and histone acetylation (H3K27ac), which marks active enhancer regions in the genome. Transcription activity in murine myocardium was increased when the tested enhancer element contained transcription factor binding regions, whereas histone acetylation had only a minor effect. For application in future gene therapy approaches of cardiac diseases, novel cardiac enhancer elements should be evaluated also in pathological situations such as heart failure or cardiac ischemia as well as in large animal models. Promising results have been reported for candidate enhancer elements of the LMNA and MYH7 genes which are expressed in the heart and upregulated in cardiomyopathy [59]. These enhancers could allow disease-induced transcription and increase transgene expression only under pathological conditions, as shown by a CRISPR-based deletion approach in human iPSCs and H1 cells. Further studies are necessary to investigate whether those candidate enhancers are suitable also for vector-based gene transfer approaches.

## 4. Cis-Regulatory Elements of the Cardiac Conduction System (CCS)

Many cardiac diseases need an even more restricted gene transfer in case of future gene therapeutic approaches. These include therapeutic targets involved in control of cardiac excitation and transduction as impaired in sick sinus syndrome, atrioventricular (AV) or bundle branch blocks, and ion channel disorders, leading to life-threatening cardiac arrythmias [60]. Delivery of light-inductive channelrhodopsin-2 via AAV9 expressed under a constitutive CAG promoter led to optogenetic pacing of rat hearts by blue-light illumination [61]. Among others, this promising therapeutic approach could be improved by restricting transgene expression to cardiomyocytes of the conduction system using either specific promoter or enhancer elements.

Transcripts of the short stature homeobox 2 (SHoX2) are specifically expressed in nodal-like cells, as indicated by screening of human-induced pluripotent stem cell-derived cardiomyocytes (hiPSC-CMs) [60]. The transcription factor SHoX2 plays a major role in the formation and differentiation of the SAN but is also involved in palate and limb development [62]. A 3.5 kb promoter element of SHoX2 drives expression of a voltage-sensitive fluorescent protein (VSFP) specifically in nodal-like hiPCS-CMs after lentiviral delivery [63]. Chen et al. thereby presented a first approach of a potential CCS-specific promoter, which in the future could be used in a gene therapeutical context.

Enhancer sequences specific to nodal cardiomyocytes were introduced by the Christoffels lab. They identified two regulatory elements of the T-box transcription factor 3-encoding (Tbx3) gene and enhancers of SCN5A/SCN10A and TBX5 [64,65]. Tbx3 expression is specific for the AV conduction system, and a heterologous loss results in AV bundle hypoplasia and restrains AV conduction, suggesting Tbx3 regulatory elements as a potential tool in gene therapy [66]. The same study identified regulatory elements correlating with Tbx3 and Tbx5 expression. In comparison to other subtypes, gene delivery specific to nodal cardiomyocytes is at a very early stage. Validation is pending of whether SHoX2 and Tbx3 and Tbx5 regulatory sequences could also mediate CCS-specific expression in an (adeno-associated) vector-based approach in vitro and in vivo.

## 5. Conclusions

Several cardiac promoters and enhancers could serve to mediate tissue-specific transgenic expression on the transcriptional level. Different promoters enable different degrees of specificity which could be advantageous for particular diseases. Upregulation of promoter activity in disease states could establish an ‘induction-by-disease’ switch. Here, transcription would be elevated in disease states and basal in physiological conditions protecting healthy tissue. Especially in the context of the limiting packaging size of AAVs, a small promoter fragment is desirable to facilitate enough space for transgene cDNA.

The MLC-2v promoter provides ventricle-restricted transcription, but expression levels are relatively low due to postnatal downregulation. Combining with CMV enhancer elements increases MLC-2v promoter activity in the cardiac ventricle and other tissues, resulting in long promoter sequences or lack of specificity when using the 260 bp MLC-2v promoter fragment. Instead of unspecific CMV enhancers, cardiac-specific enhancer elements like the CS-CRM4 provide increased expression and heart specificity. Under physiological conditions, the ANF promoter is exclusively active in atria and mediates low transgenic expression. ANF is upregulated in hypertrophied and failing hearts and extends to ventricular tissue, suggesting the ANF promoter as a promising ‘induction-by-disease’ switch. The SLN promoter also mediates atrial gene expression, but low off-target activity in skeletal muscle and diaphragm was detected. Coupling the SLN promoter to the cardio-specific enhancer module restricts transgene expression to the atria, but the hybrid-promoter size is relatively large. SLN upregulation in congenital heart disease, preserved ejection fraction, and mitral regurgitation makes the SLN promoter a potential tool for disease-induced gene expression in these pathological states. However, as for the ANF promoter, the suitability for disease-induced promoter activity has not been finally shown in the AAV context. The cTnT promoter mediates restricted gene expression in the whole myocardium, even with small promoter fragments around 500 bp, suggesting that it may also be useful with small double-stranded vector genomes. Coupling it to CS-CRM4 increases expression, while specificity remains or improves. Cardiac gene expression under the αMHC promoter can be increased via fusion to cardiac enhancers. Due to the downregulation of αMHC in heart failure states, the αMHC promoter might not be suitable for potential gene therapy in heart failure or patients with risk for heart failure but rare genetic disorders like storage diseases.

Many recent preclinical and clinical studies use AAV9 vectors due to its high cardiotropism. Table 1 indicates applications of transcriptional targeting approaches to the heart with AAVs. While promising preclinical studies reinforce the use of promotors with activity restricted to myocardial tissue in cardiac gene therapy, current clinical trials targeting cardiac diseases use unspecific, constitutively active promoters. One study aims to treat heart failure with intracoronary injection of a chimeric AAV2/AAV8 vector delivering protein phosphatase inhibitor 1 under control of the CMV promoter to block protein phosphatase 1 (NAN-101, NCT041796643). A further construct is tested in patients with Danon disease, who, among others, suffer from hypertrophic cardiomyopathy because of a defective lysosome-associated membrane protein 2 isoform B (LAMP2B). For treatment, a functional copy of the LAMP2B gene driven by the CAG promoter is delivered with an AAV9 vector through intravenous injection (NCT03882437). The use of unspecific constitutively active promoters in both recent trials indicates that tissue-specific promoters have not arrived in clinical applications yet. However, no vector harboring a cardio-specific promoter or enhancer has been tested in a human trial so far. Various ongoing and already-approved clinical trials with gene-therapy vectors targeting skeletal and cardiac muscle via muscular promoters serve as a proofs-of-concept and provide promise for a future application of cardiac-targeted AAV vectors.

## Figures and Tables

**Figure 1 pathogens-12-01301-f001:**
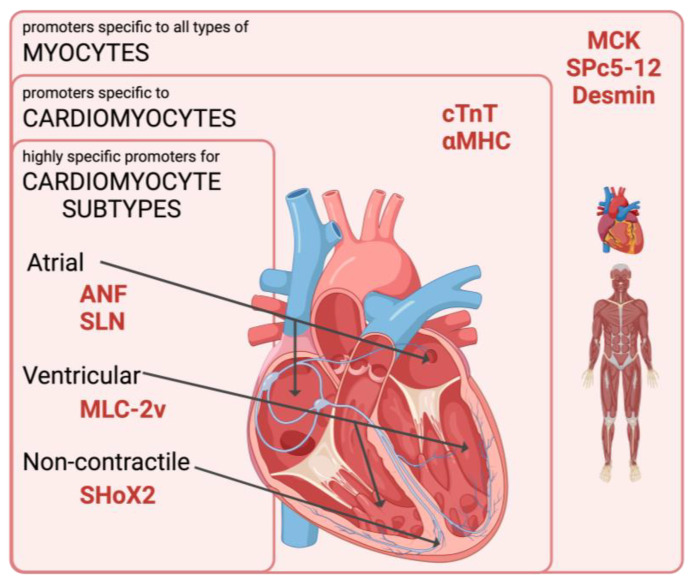
Levels of specificity are mediated by promoters in cardiac gene transfer. Specificity is low when promoters (MCK, SPc5-12, and desmin) are active in all myocytes including skeletal and cardiac muscles. The promoters cTnT and α-MHC are only active in cardiomyocytes. The highest level of specificity is mediated by cardiomyocyte-subtype-specific promoters. ANF and SLN are only expressed in atrial contractile cardiomyocytes and MLC-2v is restricted to ventricular cardiomyocytes. SHoX2 is a promoter potentially specific to non-contractile cardiomyocytes shaping the cardiac conduction system (CCS). Figure created with BioRender.com.

**Table 1 pathogens-12-01301-t001:** Cardiac promoters used in the context of AAVs.

Promoter	Specificity
MLC-2v	The 1.5 kb promoter coupled to the CMV enhancer increased luciferase expression in mice systemically delivered by AAV2; some ectopic expression in liver, lung, and skeletal muscle remained [23].
	The 1.5 kb promoter coupled to CMV enhancer in an AAV9 vector restored S100A1 expression in left ventricular myocardium of pigs after coronary venous retrofusion in a model of ischemic cardiomyopathy [11].
	A 281 bp promoter fragment specifically drives (low) GFP expression in mice delivered by AAV2 and neonatal rat myocardium [8].
	A 263 bp promoter fragment coupled to 567 bp CMV enhancer drives GFP expression in murine heart on levels comparable to CMV promoter after systemic delivery by AAV9, but ectopic expression in liver remained [3].
ANF	The 653 bp promoter mediates (low) atria specific GFP expression after systemic delivery by AAV9 in mice [33].
SLN	The 1029 bp promoter coupled to 219 bp Cas2 enhancer element systemically delivered by AAV9 increased luciferase and GFP expression in murine atria and reduces off-target expression, compared to SLN promoter only [37].
cTnT	A 418 pb chicken promoter sequence drives luciferase and GFP expression in murine myocardium best when systemically delivered by AAV9 followed by AAV8, with significantly reduced ectopic expression compared to CMV promoter; lower expression levels when delivered by AAV1, AAV2, or AAV6 [42].
	The 418 bp chicken promoter coupled to 219 bp Cas2 enhancer element drives high and stable luciferase expression in mice systemically delivered by AAV9 and further increased specificity compared to cTnT promoter alone [9].
	The chicken cTnT promoter (Addgene 105543) restored AIP expression in a mouse model for inherited cardiac arrythmia and iPCS of human patients when systemically delivered by AAV9 [43].
	A 544 bp human cTnT promoter element integrated in AAV9 vector mediates Cre recombinase expression in a LacZ reporter mouse line after systemic injection, mainly in heart with minimal off-target expression [10].
αMHC	A 363 bp truncated promoter restores gene expression specifically in cultured rat cardiomyocytes and in murine myocardium after local injection [47].
	A 363 bp promoter fragment coupled to several enhancer elements mediates specific and high myocardial expression in mice systemically delivered by AAV9; highest cardio specific expression was mediated by CS-CRM4, a 219 bp sequence of Casq2 enhancer [37].
MCK	The 770 bp MHCK7 expression cassette consists of a 565 bp core MCK promoter fragment + 50 bp sequence of the first non-coding exon coupled to modified MCK enhancer (deleted region between E-box and MEF2); TSS was replaced with a Inr consensus sequence and addition of a 188bp αMHC- enhancer element. The MHCK7 promoter mediates high-level microdytrophin expression in a mouse model of Duchenne muscular dystrophy systemically delivered by AAV6 (reduced ectopic expression remained) [52] and is already approved in a gene therapy drug (NCT03375164). MCK promoters mediate transgene expression after intravenous injection.
	The 837 bp CK8 expression cassette is like the MHCK7 promoter but contains two copies of the MCK enhancer and lacks the αMHC-enhancer element; an AAV9 vector harboring CK8 controls microdystrophin expression in a phase II trial (NCT03368742).
SPc5-12	The 334 bp SPc5-12 synthetic promoter coupled to 215 bp CS-CRM4 in a AAV9 drives luciferase expression in murine heart and skeletal muscle after systemic injection [37].
	The 334 bp SPc5-12 promoter coupled to 215 bp CS-CRM4 systemically delivered by AAV9 increases microdystrophin expression in cardiac and skeletal muscles in a mouse model for Duchenne muscular dystrophy [55].

## Data Availability

Not applicable.

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
