# Peer review of "Transcriptional Targeting Approaches in Cardiac Gene Transfer Using AAV Vectors"

_pathogens, 2023, doi:10.3390/pathogens12111301_

Round 1

Reviewer 1 Report

Comments and Suggestions for Authors

Author Response

Thank you very much for taking the time to carefully review our manuscript and giving us the valuable suggestions that helped to further improve our work. Please find the responses below.

Comments 1: Regarding paragraph 2, it should reference additional articles on cardiac-specific promoters and enhancers to provide a richer context and support the arguments presented.

Response 1: Thank you for this annotation. We included the desmin promotor in our manuscript since it works in combination with AAVs and promising clinical trials using this promoter are ongoing. The paragraph is included in page 7.

“2.7. Desmin promoter

Different combinations of promoter/enhancer sequences of the gene encoding desmin have been recently reviewed in detail [13]. The desmin promoter is active in cardiac, skeletal, and smooth muscles and in vivo studies indicated similar expression levels mediated by the desmin promoter in comparison to the highly active CMV promoter [13]. Promising preclinical studies revealed for example the construct rAAV.DES.hGAA which drives acid a-glucasidase expression under a human desmin promoter-enhancer construct packaged into the AAV9 capsid. The vector is currently tested in a clinical trial to restore acid a-glucasidase levels in patients with Pompe disease for reduction of glycogen accumulation in cardiac and skeletal muscle tissue (NCT02240407).”

Comments 2: Regarding the citations on the ANF promoter, could the most recent research be included to provide the most recent evidence in the field.

Response 2: Thank you for this comment. We included a more recent work from Hulsurkar et al., where the ANF promoter from Ni et. al was used to generate an atria-specific knockdown in a mouse model. Is was included on p. 4.

„The same vector was used to introduce an atria-specific liver kinase-B1(LKB1) knockdown in LKB1fl/fl mice by delivery of Cre [67].”

Comments 3: It is noteworthy to mention the recent FDA approval of ELEVIDYS, the first gene therapy for the treatment of Duchenne Muscular Dystrophy.

Response 3: Thank you for pointing this out. We agree and included the gene therapy in our manuscript on page 7.

“Success of clinical trials led to approval of delandistrogene moxeparvovec (ELEVIDYS), a recombinant AAVrh74 vector harboring the MHCK7 promoter element to express micro-dystrophin in pediatric Duchenne patients (NCT03375164).”

Comments 4: Could you provide details of recent clinical trials such as NCT04179643 led by Asklepios Biopharmaceutical for congestive heart failure and NCT03882437 for Danon disease? It would be particular interesting to know the types of promoters and enhancers used in these trails.

Response 4: Thank you for mentioning clinical trials targeting cardiac conditions. The vector used in NCT04179643 contains an unspecific CMV promoter and in NCT03882437 an unspecific CAG promoter was used. Unfortunately, no tissue-specific promoters were used but we included both trials in the manuscript to point out the relevance of cardiac targeted gene therapy. The trials were included on page 12.

“While promising preclinical studies reinforce the use of promotors with activity restricted to myocardial tissue in cardiac gene therapy, current clinical trials targeting cardiac diseases use unspecific, constitutively active promoters. One study aims to treat heart failure by intracoronary injection of a chimeric AAV2/AAV8 vector delivering protein phosphatase inhibitor 1 under control of the CMV promoter to block protein phosphatase 1 (NAN-101, NCT041796643). A further construct is tested in patients with Danon Disease, who among others suffer from hypertrophic cardiomyopathy because of a defective lysosome-associated membrane protein 2 isoform B (LAMP2B). For treatment, a functional copy of the LAMP2B gene driven by the CAG promoter is delivered with an AAV9 vector through intravenous injection (NCT03882437). The use of unspecific constitutively active promoters in both recent trials indicates that tissue-specific promoters have not arrived in clinical applications yet.”

Reviewer 2 Report

Comments and Suggestions for Authors

The review article authored by Schröder et al. provides a comprehensive overview of the various approaches that have been investigated to develop cardiac specific promoters to be used in AAV gene therapies targeting diseases of the heart. The manuscript is well written and informative. I have only minor comments for the authors.

1. Figure 1 is somewhat difficult to interpret, at least for this reviewer. Might there be a more straightforward/alternative way to present this information?

2. Throughout the manuscript, please indicate the route of administration used to deliver the AAV/Lenti/AdV vector containing the putative cardiac specific promoter. This information is missing from some sections.

3. Table 1 - Please include route of administration used for all studies cited in this table.

4. Remove the last sentence of the manuscript "This section is not mandatory but can be added to the manuscript if the discussion is unusually long or complex. "

Author Response

Thank you very much for taking the time to carefully review our manuscript and giving us the valuable suggestions that helped to further improve our work. Please find the responses below.

Comments 1: Figure 1 is somewhat difficult to interpret, at least for this reviewer. Might there be a more straightforward/alternative way to present it.

Response 1: Thank you for this comment. We adapted the figure to point out that the promotors are active in all myocytes including cardiomyocytes and still in different types of cardiomyocytes. The next smaller box includes promoters active in cardiomyocytes with activity in all subtypes. The smallest box includes the most specific promoters in the muscular context that are specific to the subtypes (figure was uploaded).

Comments 2: Throughout the manuscript, please indicate the route of administration used to deliver the AAV/Lenti/AdV vector containing the putative cardiac specific promoter. This information is missing from some sections.

Response 2: Thank you for pointing this out. This is indeed very important, and we added the administration to all reviewed studies.

Comments 3: Table-1 Please include route of administration used for all studies cited in this table.

Response 3: The administration route was also added in the table.

Comments 4: Remove the last sentence on the manuscript: “This section is not mandatory but can be added to the manuscript if the discussion is usually long or complex.”

Response 4: Thank you for this annotation. Of course we removed that sentence.
